# Dynamic Resolution Network

**Mingjian Zhu**[1,2,4,5*], **Kai Han**[2,3*] **Enhua Wu**[3,6], **Qiulin Zhang**[7], **Ying Nie**[2],
**Zhenzhong Lan**[4,5], **Yunhe Wang**[2†]
[1]Zhejiang University.  [2]Huawei Noah's Ark Lab.
[3]State Key Lab of Computer Science, ISCAS & University of Chinese Academy of Sciences.
[4]School of Engineering, Westlake University.
[5]Institute of Advanced Technology, Westlake Institute for Advanced Study.
[6]University of Macau.  [7]BUPT.
zhumingjian@zju.edu.cn, {hankai,weh}@ios.ac.cn,
lanzhenzhong@westlake.edu.cn, yunhe.wang@huawei.com

## Abstract

Deep convolutional neural networks (CNNs) are often of sophisticated design with numerous learnable parameters for the accuracy reason. To alleviate the expensive costs of deploying them on mobile devices, recent works have made huge efforts for excavating redundancy in pre-defined architectures. Nevertheless, the redundancy on the input resolution of modern CNNs has not been fully investigated, *i.e.*, the resolution of input image is fixed. In this paper, we observe that the smallest resolution for accurately predicting the given image is different using the same neural network. To this end, we propose a novel dynamic-resolution network (DRNet) in which the input resolution is determined dynamically based on each input sample. Wherein, a resolution predictor with negligible computational costs is explored and optimized jointly with the desired network. Specifically, the predictor learns the smallest resolution that can retain and even exceed the original recognition accuracy for each image. During the inference, each input image will be resized to its predicted resolution for minimizing the overall computation burden. We then conduct extensive experiments on several benchmark networks and datasets. The results show that our DRNet can be embedded in any off-the-shelf network architecture to obtain a considerable reduction in computational complexity. For instance, DR-ResNet-50 achieves similar performance with an about 34% computation reduction, while gaining 1.4% accuracy increase with 10% computation reduction compared to the original ResNet-50 on ImageNet. Code will be available at https://gitee.com/mindspore/models/tree/master/research/cv/DRNet.

## 1 Introduction

Deep convolutional neural networks (CNNs) have achieved remarkable success in various computer vision tasks, under the development of algorithms [6, 22, 37], computation power, and large-scale datasets [2, 17]. However, the outstanding performance is accompanied by large computational costs, which makes CNNs difficult to deploy on mobile devices. With the increasing demand for CNNs on real-world applications, it is imperative to reduce the computational cost and meanwhile maintain the performance of neural networks.

Recently, researchers have devoted much effort to model compression and acceleration methods, including network pruning, low-bit quantization, knowledge distillation, and efficient model design.

---

*Equal contribution. † Corresponding author. This research has been supported by the Key R&D program of Zhejiang Province (Grant No. 2021C03139). This work was supported by NSFC (62072449, 61632003), Guangdong-Hongkong-Macao Joint Research Grant (2020B1515130004), and Macao FDCT (0018/2019/ AKP).

Network pruning aims to prune the unimportant filters or blocks that are insensitive to model performance through a certain criterion [31, 15, 18, 25]. Low-bit quantization methods represent weights and activation in neural networks with low-bit values [11, 13]. Knowledge distillation transfers the knowledge of the teacher models to the student models to improve the performance [7, 38, 40]. The efficient model design utilizes lightweight operations like depth-wise convolution to construct some novel architectures successfully[8, 42, 5]. Orthogonal to those methods that usually focus on the network weights or architectures, Guo *et.al.* [4] and Wang *et.al.* [33] study the redundacy that exists in the input images. However, the resolutions of the input images in most of existing compressed networks are still fixed. Although deep networks are often trained using an uniform resolution (*e.g.*, 224× 224 on the ImageNet), sizes and locations of objects in images are radically different. Figure 1 shows some samples that the required resolution for achieving the highest performance are different. For the given network architecture, the FLOPs (floating-number operations) of the network for processing image will be significantly reduced for images with lower resolution.

Admittedly, the input resolution is a very important factor that affects the computational costs and the performance of CNNs. For the same network, a higher resolution usually results in larger FLOPs and higher accuracy [26]. In contrast, the model with a smaller input resolution has lower performance while the required FLOPs are also smaller. However, the shrink of input resolutions of deep networks provides us another potential to alleviate the computation burden of CNNs. To have an explict illustration, we first test some images under different resolutions with a pre-trained ResNet50 as shown in Figure 1 and count the minimum resolution required to give the correct prediction for each sample. In practice, "easy" samples, such as the panda with obvious foreground, can be classified correctly in both low and high resolution, and "hard" samples, such as the damselfly whose foreground and background are tangled can only be predicted accurately in high resolution. This observation indicates that a larger proportion of images in our datasets can be efficiently processed by reducing their resolutions. On the other hand, it is also compatible with the human perception system [1], *i.e.*, some samples can be

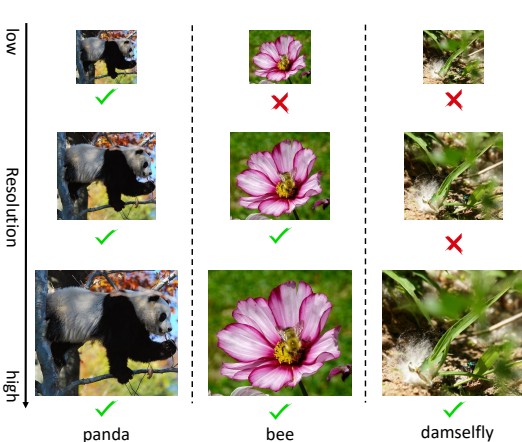

Figure 1: The prediction results of a well-trained ResNet-50 model for samples under different resolutions (112×112, 168×168, 224×224). Some "easy" samples like the left column (panda), can be classified correctly using both low and high resolutions. However, some "hard" samples like the right column (damselfly), where the foreground objects are hidden or blend with the background, can only be classified correctly using the high resolution.

understood easily just in blurry mode while the others need to be seen in clear mode.

In this paper, we propose a novel dynamic-resolution network (DRNet) which dynamically adjusts the input resolution of each sample for efficient inference. To accurately find the required minimum resolution of each image, we introduce a resolution predictor which is embedded in front of the entire network. In practice, we set several different resolutions as candidates and feed the image into the resolution predictor to produce a probability distribution over candidate resolutions as the output. The network architecture of the resolution predictor is carefully designed with negligible computational complexity and trained jointly with classifier for recognition in an end-to-end fashion. By exploiting the proposed dynamic resolution network inference approach, we can excavate the redundancy of each image from its input resolution. Thus, computational costs of easy samples with lower resolutions can be saved, and the accuracy for hard samples can also be preserved by maintaining higher resolutions. Extensive experiments on the large-scale visual benchmarks and the conventional ResNet architectures demonstrate the effectiveness of our proposed method for reducing the overall computational costs with comparable network performance.

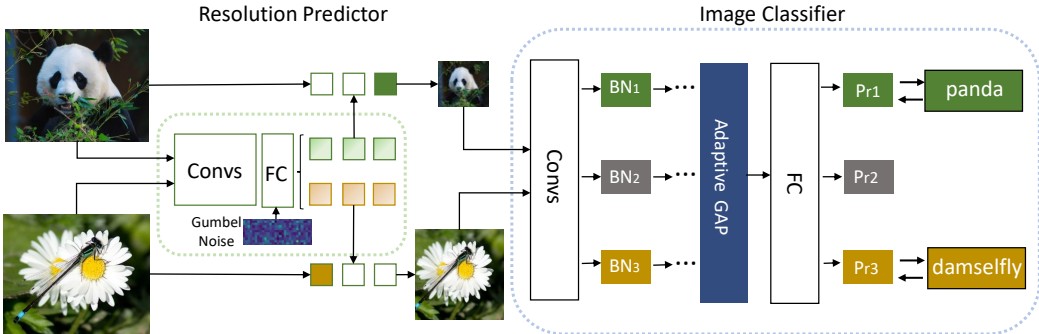

Figure 2: Overall framework, the resolution predictor guides the resolution selection for the large classifier. 'BN': batch normalization layer; '$P_{ri}$': probability distribution over categories under resolution $ri$; In the inference stage, a one-hot vector is predicted by the resolution predictor, in which the '1' denotes a corresponding selected resolution. The original image is then resized to the selected resolution and input to the large classifier with chosen BN.

## 2   Related Works

Although deep CNN models have shown excellent accuracy, they often contain millions of parameters and FLOPs. Thus the model compression techniques are becoming a research hostpot for reducing the computational costs of CNNs. Here, we revisit existing works in two parts, *i.e.*, static model compression and dynamic model compression.

We classify model compression methods that are not instance-aware as the static. Group-wise Convolution (GWC), Depth-wise Convolution and Point-wise Convolution are widely used for efficient model design, such as MobileNet [8], ResNeXt [36], and ShuffleNet [42]. Revealing pattern redundancy among feature maps, Han [5] proposes to generate more feature maps from intrinsic ones through some cheap operations and Zhang *et.al.* [41] adopts relatively heavy computation to extract intrinsic information while tiny hidden details are processed with some light-weight operations. The methods above achieve model acceleration to some extent. However, they treat all input samples equally, whereas the difficulty for CNN models or humans to recognize each sample is unequal. So instance-aware model compression can be further explored.

Dynamic model compression takes the unequal difficulty of each sample into consideration. Huang [9] proposes multi-scale dense networks with multiple classifiers to allocate uneven computation across "easier" and "harder" inputs. Wu *et.al.* [35] introduces BlockDrop which learns to dynamically execute the necessary layers so as to best reduce total computation. Veit *et.al.* [28] proposes ConvNet-AIG to adaptively define their network topology conditioned on the input image. Except for dynamic adjustment on model architectures, recent works pay more attention to input images. Verelst *et.al.* [29] proposes a small gating network to predict pixel-wise masks determining the locations where dynamic convolutions are evaluated. Uzkent *et.al.* [27] proposes PatchDrop to dynamically identify when and where to use high-resolution data conditioned on the paired low-resolution images. Yang *et.al.* [39] dynamically utilize sub-networks of the base network to process images with different resolutions. Wang *et.al.* [33] proposes GFNet with patch proposal networks that strategically crop the minimal image regions to obtain reliable predictions. There exists many works on multiple resolutions and dynamic mechanisms. ELASTIC [30] uses different scaling policies for different instances and it learns from the data how to select the best policy. Hydranets [20] chooses different branches for different inputs by a gate and aggregates their outputs with a combiner. RS-Net [32] utilizes private BNs, shared convolutions, and fully-connected layer and to train input images with different resolutions.

Different from dynamic adjustment on model architectures and dynamic modification on input images with reinforcement learning, we consider the whole image and propose a resolution predictor to dynamically choose the performance-sufficient and cost-efficient resolution for a single model to obtain reliable prediction with end-to-end training.

# 3 Approach

In this section, we first introduce the overall framework of the proposed Dynamic Resolution Network (DRNet), then describe the resolution predictor, resolution-aware BN, and optimization algorithm in detail, respectively.

## 3.1 Dynamic Resolution Network

Inspired by the fact that different sample requires different resolution to achieve the least accurate prediction, we propose to develop an instance-aware resolution selection approach for a single large classifier network. As shown in Figure 2, the proposed method mainly consists of two components. The first is the large classifier network with both high performance and expensive computational costs, such as the classical ResNet [6] and EfficientNet [24]. The other is a resolution predictor for finding the minimal resolution so that we can adjust the input resolution of each image to have a better trade-off on the accuracy and efficiency. For an arbitrary input image, we first forecast its suitable resolution $r$ using the resolution predictor. Then, the large classifier will take the resized image as inputs and the required FLOPs will be reduced significantly when $r$ is lower than that of the origional resolution. To achieve better performance, the resolution predictor and the base network are optimized end-to-end during training.

**Resolution Predictor.** The resolution predictor is designed as a CNN-based preprocessing operation before input samples are fed to the large base network. It's from the inspiration that our well-trained large models can also predict a relative amount of samples correctly though they are in relatively small resolutions while large amounts of computation cost can be saved. On the one hand, the goal of the resolution predictor is to find an appropriate instance-aware resolution by inferring a probability distribution over candidate resolutions. Note that there are a vast number of candidates from $1\times1$ to $224\times224$, which makes it difficult, also meaningless, for the resolution predictor to explore such a long-range of resolutions. As a simplification strategy and practical requirement, we choose $m$ resolution candidates $r_1, r_2, \cdots, r_m$ to shrink the exploration range. On the other hand, we have to keep the model size of the proposed resolution predictor as small as possible since it will bring extra FLOPs, otherwise, it becomes impractical to implement such a module if its extra introduced computation exceeds the saved one from the low resolution. In this spirit, we design the resolution predictor with a few convolutional layers and fully-connected layers to complete a resolution classification task. Then the preprocessing of the proposed resolution predictor $R(\cdot)$ can be given as follows:

$$p_r = [p_{r_1}, p_{r_2}, ..., p_{r_m}] = R(X), \tag{1}$$

where $X$ is the input samples fed to the resolution predictor, $m$ is the total number of candidate resolutions, and $p_r \in \mathbb{R}^m$ is the outputs of the resolution predictor which represents the probability of each candidate. Then the resolution corresponding to the highest probability entry is selected as the resolution fed to the large classifier. Since the process from the soft outputs of resolution predictor to the discrete resolution resizing operation does not support end-to-end training, here we adopt Gumbel-Softmax [14] module $\mathbb{G}$ to turn soft decisions $p_r$ into hard decisions $h \in \{0,1\}^m$ by applying Gumbel-Softmax trick to solve the non-differentiable problem:

$$h = \mathbb{G}(p_r) = \mathbb{G}(R(X)), \tag{2}$$

where the Gumbel-Softmax trick will be described in the next subsection. For validation, the resolution predictor makes decisions first then the input with the selected resolution only is fed to the large classifier in the normal way as shown in the left part of Figure 2.

**Resolution-aware BN.** Our framework is proposed to use only a single large classifier for the sake of storage pressure and loading latency, which results in that the single classifier has to process multi-resolution inputs and raises two problems. One obvious problem is that the first fully-connected layer will fail to work with a different input resolution and can be solved with global average pooling. Thus we can process multiple resolutions in one single network. The other hidden problem exists in Batch Normalization (BN) [12] layers. BN is used to make deep models converge faster and more stable through channel-wise normalization of the input layer by re-centering and re-scaling. However, activation statistics including means and variances under different resolutions are incompatible [26]. Using shared BNs under multiple resolutions leads to lower accuracy in our experiments as shown

in section 4.3. Since the batch normalization layer contains a negligible amount of parameters, we propose resolution-aware BNs as shown in Figure 2. We decouple the BN for each resolution and choose the corresponding BN layer to normalize the features:

$$x_j = \gamma_j \frac{x_j - \mu_j}{\sqrt{\sigma_j{}^2 + \epsilon}} + \beta_j, \; j \in \{1, 2, ..., m\}, \tag{3}$$

where $\epsilon$ is a small number for numerical stability, $\mu_i$ and $\sigma_i$ are private averaged mean and variance from the activation statistics under separate resolutions; $\beta_i$ and $\gamma_i$ are private learnable scale weights and bias. Since shared convolutions are insensitive to performance, the overall adjustment for the original large classifier is shown in the right part of Figure 2.

## 3.2 Optimization

The proposed framework is optimized to perform instance-aware resolution selection for inputs of a single large classifier with end-to-end training. The loss function and Gumbel softmax trick are described in the following.

**Loss Function.** The base classifier and the resolution predictor are optimized jointly. The loss function includes two parts: the cross-entropy loss for image classification and a FLOPs constraint regularization to restrict the computation budget.

Given a pretrained base image classifier $\mathcal{F}$ which takes image $X$ as input and outputs the probability predictions $y = \mathcal{F}(X)$ for image classification, we optimize the resolution predictor and finetune the pretrained base classifier together so as to make them compatible with each other. For the input image $X$, we first resize it into $m$ candidate resolutions as $X_{r_1}, X_{r_2}, \cdots, X_{r_m}$. We use the proposed resolution predictor to produce the resolution probability vector $p_r \in \mathbb{R}^m$ for each image. The soft resolution probability $p_r$ is transformed into hard one-hot selection $h \in \{0, 1\}^m$ using Gumbel-Softmax trick as equation 2 where the hot entry of $h$ represents the resolution choice for each sample. We first obtain the final prediction of each resolution $y_{rj} = \mathcal{F}(X_{rj})$, and then sum them up with $h$ to obtain the recognition prediction for the selected resolution:

$$\hat{y} = \sum_{j=1}^{m} h_j y_{r_j}. \tag{4}$$

The Cross-Entropy loss $\mathcal{H}$ is performed between $\hat{y}$ and target label $y$ as follows:

$$L_{ce} = \mathcal{H}(\hat{y}, y). \tag{5}$$

The gradients from the loss $L_{ce}$ are back-propagated to both the base classifier and the resolution predictor for optimization.

If we use the Cross-Entropy loss only, the resolution predictor will converge to a sub-optimal point and tend to select the largest resolution because samples with the largest resolution correspond to relatively lower classification loss generally. Although the classification confidence of the low-resolution image is relatively lower, the prediction can be correct and requires fewer FLOPs. In order to reduce the computational cost and balance the different resolution selection, we propose a FLOPs constraint regularization to guide the learning of resolution predictor:

$$F = \sum_{j=1}^{m} (C_j \cdot h_j), \tag{6}$$

$$L_{reg} = \max \left( 0, \frac{\mathbb{E}(F) - \alpha}{C_{max} - C_{min}} \right), \tag{7}$$

where $F$ is the actual inference FLOPs, $C_j$ is the pre-computed FLOPs value for the $j$-th resolution, $\mathbb{E}(\cdot)$ is the expectation value over samples, and $\alpha$ is the target FLOPs. Through this regularization, there will be a penalty if averaged FLOPs value is too large, enforcing the proposed resolution predictor to be instance-aware and predict the resolution that is both performance-sufficient (with correct prediction) and cost-efficient (with low resolution).

Finally, the overall loss is the weighted summation over the classification loss and the FLOPs constraint regularization term:

$$L = L_{ce} + \eta L_{reg}, \tag{8}$$

where $\eta$ is a hyper-parameter to match the magnitude of $L_{ce}$ and $L_{reg}$.

**Gumbel Softmax Trick.**    Since there exists an non-differentiable problem in the process from the resolution predictor's continues outputs to discrete resolution selection, we adopt Gumbel Softmax trick [19, 14] to make discrete decision differentiable during the back-propagation. In Eq. 1, the resolution predictor gives the probabilities for the resolution candidates $p_r = [p_{r_1}, p_{r_2}, ..., p_{r_m}]$. Then the discrete candidate resolution selections can be drawn using:

$$h = \text{one\_hot}[(\arg\max_j(\log p_{r_j} + g_j)],\tag{9}$$

where $g_j$ is Gumbel noise obtained through two $\log$ operation applied on i.i.d samples $u$ drawn from a uniform distribution as follows:

$$g_j = -\log(-\log u), \quad u \sim U(0, 1).\tag{10}$$

During training, the derivative of the one-hot operation is approximated by Gumbel softmax function which is both continuous and differentiable:

$$h_j = \frac{\exp\left(\log(\pi_j) + g_j\right)/\tau}{\sum_{j=1}^m exp((\log(\pi_j) + g_j/\tau))},\tag{11}$$

where $\tau$ is the temperature parameter. The introduction of Gumbel noise has two positive effects. On the one hand, it will not influence the highest entry of the original categorical probability distribution. On the other hand, it makes the gradient approximation from discrete hardmax to continuous softmax more fluent. By this straight-through Gumbel softmax trick, we can optimize the overall framework end-to-end.

## 4   Experiments

To show the effectiveness of our proposed method, in this section, we conduct experiments on the small-scale ImageNet-100 and large-scale ImageNet-1K [2] with classic large classifier networks, including ResNet [6] and MobileNetV2 [23], where we replace their single batch normalization layer(BN) with resolution-aware BNs and add the proposed resolution predictor to guide the resolution selection.

### 4.1   Implementation Details

**Datasets.**    *ImageNet-1K* dataset (ImageNet ILSVRC2012) [2] is a widely-used benchmark to evaluate the classification performance of neural networks, which consists of 1.28M training images and 50K validation images in 1K categories. *ImageNet-100* is a subset of the ImageNet ILSVRC2012, whose training set is random selected from the original training set and consists of 500 instances of 100 categories. The validation set is the corresponding 100 categories of the original validation set. The categories of ImageNet-100 is provided in supplementary materials. For the license of ImageNet dataset, please refer to `http://www.image-net.org/download`.

**Experimental Settings.**    For data augmentation during training for both ImageNet-100 and ImageNet-1K, we follow the scheme as in [6] including randomly cropping a patch from the input image and resizing to candidate resolutions with the bilinear interpolation followed by random horizontal flipping with probability 0.5. For data processing during validation, we first resize the input image into $256 \times 256$ and then crop the center $224 \times 224$ part. The details of the resolution predictor are provided in supplementary materials. For both datasets, we firstly employ the images of different resolutions to pre-train a model without the resolution predictor. The losses of each resolution are summed up for optimization. Then we add a designed predictor to the model and conduct finetuning. Optimization is performed using SGD (mini-batch stochastic gradient descent) and learning rate warmup is applied for the first 3 epochs. In the pretraining stage, the model is trained with total epochs 70, batch-size 256, weight decay 0.0001, momentum 0.9, initial learning rate 0.1 which decays a factor of 10 every 20 epochs. We adopt a similar training scheme in finetuning stage. The total epochs are 100 with the learning rate decaying a factor of 10 every 30 epochs. We adopt $1\times$ learning rate to finetune the large classifier and $0.1\times$ learning rate to train the resolution predictor from scratch. The framework is implemented in Pytorch [21] on NVIDIA Tesla V100 GPUs.

## 4.2 ImageNet-100 Experiments

We conduct small-scale experiments on ImageNet-100 to guide the resolution selection for large classifiers ResNet-50. The resolution predictor is designed as a 4-stage residual network with input resolution $128 \times 128$ where each stage contains one residual basic block, which consumes about 300 million FLOPs. For candidate resolutions of the large classifier, we choose resolutions of $[224 \times 224, 168 \times 168, 112 \times 112]$ and we denote them as [224, 168, 112] for simplicity. Thus the last fully-connected layers of the resolution predictor contain three neurons. We only replace each batch normalization layer in ResNet-50 with three optional resolution-aware batch normalization layers, change the last average pooling layer to an adaptive average pooling layer, and then integrate the resolution predictor to form the overall framework. Experiment results are shown in Table 1.

For the calculation of the average FLOPs in Table 1, we sum up the FLOPs of each sample under the predicted resolution, take the extra FLOPs introduced by the resolution predictor into account and finally take the average over the whole validation set. From the results in Table 1, we can see our dynamic-resolution ResNet-50 obtains about 17% reduction of average FLOPs while gains 4.0% accuracy increase with the candidate resolutions [224, 168, 112]. When we tune the hyperparameters (*i.e.*, $\eta$ and $\alpha$) in FLOPs constraint regularization, the dynamic-resolution ResNet-50 obtains about 32% FLOPs reduction and achieve 1.8% increase in accuracy. We also extend the range of resolutions (*i.e.*, [224, 192, 168, 112, 96]) for fully exploration, especially the lower resolution. We can see that our DRNet still performs better than the baseline model. Setting a larger $\alpha$ can even obtain 44% FLOPs reduction with performance increase, which is shown in Table 2.

| Resolutions | RA-BN | FLOPs | Acc |
|---|---|---|---|
| [224] | - | 4.1 G | 78.5% |
| [224, 168, 112] | Yes | 3.4 G | 82.5% |
| [224, 168, 112] | Yes | 2.8 G | 81.4% |
| [224, 168, 112] | No | 2.9 G | 80.3% |
| [224, 192, 168, 112, 96] | Yes | 3.0 G | 81.9% |

Table 1: Results of ResNet-50 on ImageNet-100. The first row demonstrates the results of ResNet-50 backbones. The second row presents the results of the DRNet with regularization. The third row presents the DRNet trained with $\eta = 0.2$ and $\alpha = 2.5$ in the FLOPs constraint regularization, and the fourth row shows the DRNet w/o resolution-aware BN.

| $\eta$ | $\alpha$ | Average FLOPs | Acc |
|---|---|---|---|
| 0.2 | 2.0 | 2.3 G | 80.6% |
| 0.2 | 2.5 | 2.8 G | 81.4% |
| 0.2 | 3.0 | 3.3 G | 82.3% |
| 0.2 | 3.5 | 3.5 G | 82.6% |
| 0.1 | 3.0 | 3.3 G | 82.1% |
| 0.2 | 3.0 | 3.3 G | 82.3% |
| 0.5 | 3.0 | 3.1 G | 81.9% |
| 1 | 3.0 | 3.1 G | 81.5% |

Table 2: Influence of FLOPs Constraint Regularization.

## 4.3 Ablation Study

To form the dynamic resolution network, we propose two adjustments, 1) replacing each BN layer with resolution-aware BNs; 2) proposing a FLOPs balance regularizer. Here we conduct ablation studies on ImageNet-100 to investigate the influence of each part.

**Resolution-Aware BN.** Here we compare the results where the large classifier ResNet-50 is equipped with resolution-aware BN or not. From Table 1, we can see that resolution-aware BNs obtain extra one more points for ResNet-50 with similar computational cost, which demonstrates that we need to normalize feature maps with different resolutions separately, thus their activation statistics can be more accurate.

**Influence of FLOPs Constraint Regularization.** Here we explore the influence of the penalty factor $\eta$ and target FLOPs value $\alpha$ in the FLOPs constraint regularization as shown in Table 2. We first fix $\eta$ as 0.2 and tune $\alpha$ from 2.0 to 3.5. We can see that the average FLOPs increase from 2.3G to 3.5G gradually, and the accuracy also increase consequently. As for $\eta$, we fix $\alpha = 3.0$ and tune $\eta$ in the range of [0.1, 1]. A larger penalty factor leads to lower FLOPs and accuracy. That is to say when selecting images dynamically, the resolution predictor with lower penalty $\eta$ tends to choose the larger resolution, where the effect of the balance regularizer is relatively weaker.

## 4.4 ImageNet-1K Experiments

**ResNet Results.** We conduct large-scale experiments with DR-ResNet-50 on ImageNet-1K as shown in Table 3. When we set the candidate resolutions as [224, 168, 112], our DR-ResNet-50 also outperforms the baseline by 1.4 percentage with 10% FLOPs reduced. Similar to the results in Table 2, the effectiveness of FLOPs constraint regularization is also verified in Table 3, where the FLOPs drop with larger $\alpha$. Our DRNet focuses on input resolution and keeps the structure of the large classifier almost unchanged, thus the parameters of our DRNet-equipped models are more than the original large classifier due to the introduction of the resolution predictor. In other words, since our DRNet is orthogonal to those architecture compression methods, careful combinations of the two methods would make a more compact result.

| Model | $\alpha$ | Params | FLOPs | ↓FLOPs | Acc@1 | Acc@5 |
|---|---|---|---|---|---|---|
| ResNet-50-baseline | - | 25.6 M | 4.1 G | - | 76.1% | 92.9% |
| DR-ResNet-50 | - | 30.5 M | 3.7 G | 10% | 77.5% | 93.5% |
| DR-ResNet-50 | 2.0 | 30.5 M | 2.3 G | 44% | 75.3% | 92.2% |
| DR-ResNet-50 | 2.5 | 30.5 M | 2.7 G | 34% | 76.2% | 92.8% |
| DR-ResNet-50 | 3.0 | 30.5 M | 3.2 G | 22% | 77.0% | 93.2% |
| DR-ResNet-50 | 3.5 | 30.5 M | 3.7 G | 10% | 77.4% | 93.5% |
| ResNet-101-baseline | - | 44.5 M | 7.8 G | - | 77.4% | 93.5% |
| DR-ResNet-101 | - | 49.4 M | 7.0 G | 10% | 79.0% | 94.3% |

Table 3: ResNet-50 and ResNet-101 results on the ImageNet-1K dataset.

We also compare DR-ResNet-50 with other representative model compression methods to verify the superiority of the proposed method. The compared methods include Sparse Structure Selection (SSS) [10], Versatile Filters [34], PFP [16], and C-SGD [3]. As shown in Table 4, our DR-ResNet-50 achieves better performance than other methods with similar FLOPs.

| Model | Params | FLOPs | ↓FLOPs | Acc@1 | Acc@5 |
|---|---|---|---|---|---|
| ResNet-50-baseline | 25.6 M | 4.1 G | - | 76.1% | 92.9% |
| ResNet-50 (192×192) | 25.6 M | 3.0 G | 27% | 74.3% | 91.9% |
| SSS-ResNet-50 [10] | - | 2.8 G | 32% | 74.2% | 91.9% |
| Versatile-ResNet-50 [34] | 11.0 M | 3.0 G | 27% | 74.5% | 91.8% |
| PFP-A-ResNet-50 [16] | 20.9 M | 3.7 G | 10% | 75.9% | 92.8% |
| C-SGD70-ResNet-50 [3] | - | 2.6 G | 37% | 75.3% | 92.5% |
| RANet [39] | - | 2.3 G | 44% | 74.0% | - |
| DR-ResNet-50 | 30.5 M | 3.7 G | 10% | 77.5% | 93.5% |
| DR-ResNet-50 ($\alpha = 2.0$) | 30.5 M | 2.3 G | 44% | 75.3% | 92.2% |

Table 4: Comparison with other model compression methods on the ImageNet-1K dataset.

**Effect of Dynamic Resolution.** To evaluate the effect of the proposed dynamic resolution mechanism, we compare DR-ResNet-50 with randomly selected resolution. We repeat the random selection 3 times and report their accuracies and FLOPs in Imagenet-1K dataset. From the results in Table 5, DRNet shows much better performance than random baseline, indicating the effectiveness of dynamic resolution.

**On-device Acceleration of Dynamic Resolution.** In Figure 3, we demonstrate the practical accelerations of our DR-ResNet-50, which are obtained by measuring the forward time on a Intel(R) Xeon(R) Gold 6151 CPU. We directly set the batch size as 1 and the input resolution for the resolution predictor is 128. The candidate resolutions are 224, 168, and 112. We average the test time in ImageNet-1K val set. Our model substantially outperforms ResNet-50 by a significant margin.

**MobileNetV2 Results.** We also test our method on a representative lightweight neural network, *i.e.* MobileNetV2 [23]. We set the candidate resolution as [224, 168, 112]. The training setting of MobileNetV2 follows that in the original paper [23] for a fair comparison. To reduce the FLOPs

| Model | FLOPs | Acc (%) |
|---|---|---|
| Random-1 | 2.6 G | 74.70 |
| Random-2 | 2.6 G | 74.65 |
| Random-3 | 2.6 G | 74.60 |
| Random (mean) | 2.6 G | 74.65±0.04 |
| DRNet | 2.7 G | 76.2 |

Table 5: Dynamic resolution vs. Random resolution.

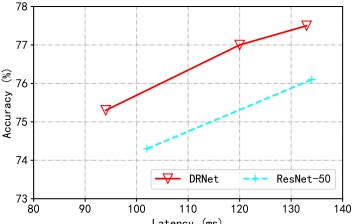

Figure 3: Acc *vs.* Latency.

of the resolution predictor, we replace the residual block with an inverted residual block and set the input size as 64×64. From the results in Table 6, we can see that DRNet achieves 72.7% top-1 accuracy with fewer computational costs.

| Model | Params | FLOPs | ↓FLOPs | Acc@1 |
|---|---|---|---|---|
| MobileNetV2-baseline | 3.5 M | 300 M | - | 71.8% |
| MobileNetV2 (192×192) | 3.5 M | 221 M | 26% | 70.7% |
| MobileNetV2-0.75× | 2.6 M | 209 M | 30% | 69.8% |
| DR-MobileNetV2 | 3.8 M | 268 M | 10% | 72.7% |

Table 6: MobileNetV2 results on the ImageNet-1K dataset.

## 4.5 Visualization

The prediction results of the resolution predictor are visualized in Figure 4. The first four samples with obvious foreground which occupy most of the whole image are predicted to use $112 \times 112$ resolution in high confidence. The middle three whose foreground is a little blurred are predicted to select $168 \times 168$ resolution. The last three samples' hidden foregrounds nearly blend with the background, thus the largest resolution are selected. Although the "easy" and "hard" examples may be different for humans and machines, these results are compatible with the human perception system.

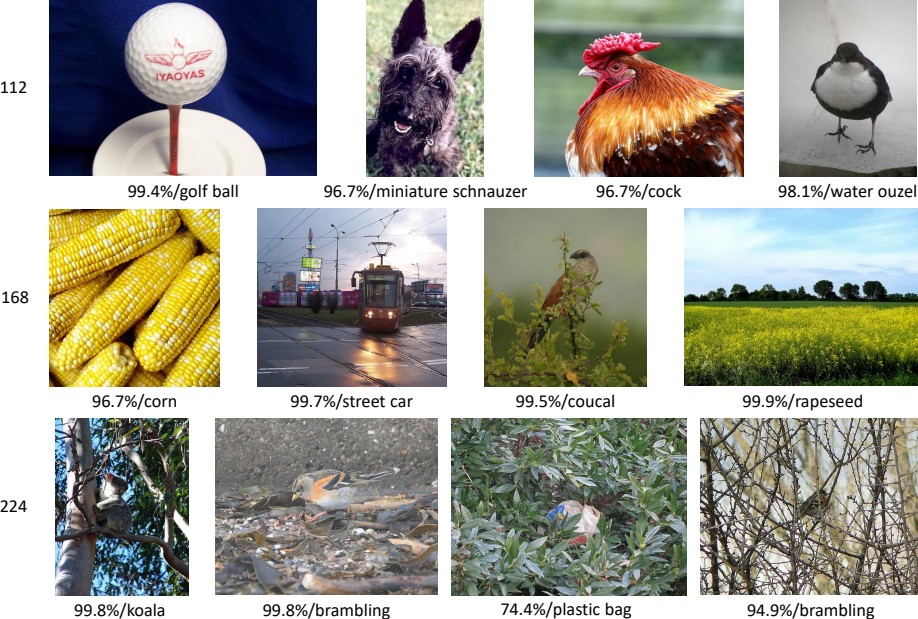

Figure 4: Image visualization results of DR-ResNet-50. Each row denotes the selected resolutions for these images. Image classification confidences and labels are shown below the images.

# 5   Conclusion

In this paper, we reveal that different sample acquires different resolution threshold to achieve the least accurate prediction. Thus large amounts of computation cost can be saved for some easier samples under lower resolutions. To make CNNs predict efficiently, we propose a novel dynamic resolution network to dynamically choose the performance-sufficient and cost-efficient resolution for each input sample. Then the input is resized to the predicted resolution and fed to the original large classifier, in which we replace each BN layer with resolution-aware BNs to accommodate the multi-resolution input. The proposed method is decoupled with the network architecture and can be generalized to any network. Extensive experiments on various networks demonstrate the effectiveness of DRNet.

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
