# Dynamic Resolution Network (Supplementary Material)

**Mingjian Zhu**[1,2,4,5*], **Kai Han**[2,3*], **Enhua Wu**[3,6], **Qiulin Zhang**[7], **Ying Nie**[2],
**Zhenzhong Lan**[4,5], **Yunhe Wang**[2†]
[1]Zhejiang University. [2]Huawei Noah's Ark Lab.
[3]State Key Lab of Computer Science, ISCAS & University of Chinese Academy of Sciences.
[4]School of Engineering, Westlake University.
[5]Institute of Advanced Technology, Westlake Institute for Advanced Study.
[6]University of Macau. [7]BUPT.
zhumingjian@zju.edu.cn, {hankai,weh}@ios.ac.cn,
lanzhenzhong@westlake.edu.cn, yunhe.wang@huawei.com

## 1 Details of predictor architectures

We utilize the basic block of resnet to construct the predictor, i.e., 4 basic blocks are stacked to form the predictor network in our paper. Here we build 4 predictor architectures with fewer parameters and FLOPs and we compare their performances as follows. We can see that the predictor with fewer flops leads to slight accuracy degradation.

(1) Predictor-Architecture-1: The original predictor in our paper. The parameters of the first convolution are Conv2d (3, 64, kernel_size=(7, 7), stride=(2, 2), padding=(3, 3), bias=False).

(2) Predictor-Architecture-2: We reduce the blocks of the predictor in (1) to construct a new predictor. We retain only 2 blocks.

(3) Predictor-Architecture-3: We increase the stride of the first convolution in (2) to 4. Thus, the parameters of the first convolution are Conv2d (3, 64, kernel_size=(7, 7), stride=(4, 4), padding=(3, 3), bias=False).

(4) Predictor-Architecture-4: We construct the predictor with only two convolutions.

| Predictor | Predictor FLOPs | Total FLOPs | Acc |
|-----------|-----------------|-------------|-------|
| (1) | 0.29G | 3.35G | 82.5% |
| (2) | 0.17G | 3.23G | 82.0% |
| (3) | 0.04G | 3.10G | 82.0% |
| (4) | 0.09G | 3.15G | 81.4% |

Table 1: comparison of different Predictors.

The details of these predictors are shown as follows:

(1) Predictor-Architecture-1

```
1  ResNet(
   (conv1)
3  (bn1)
```

*Equal contribution. † Corresponding author. This research has been supported by the Key R&D program of Zhejiang Province (Grant No. 2021C03139). This work was supported by NSFC (62072449, 61632003), Guangdong-Hongkong-Macao Joint Research Grant (2020B1515130004), and Macao FDCT (0018/2019/ AKP).

35th Conference on Neural Information Processing Systems (NeurIPS 2021).

```
  ( relu )
5 ( maxpool )
  ( layer1 ): Sequential(
7 (0): BasicBlock(
  ( conv1 )
9 ( bn1 )
  ( relu )
11 ( conv2 )
  ( bn2 )
13 )
  )
15 ( layer2 ): Sequential(
  (0): BasicBlock(
17 ( conv1 )
  ( bn1 )
19 ( relu )
  ( conv2 )
21 ( bn2 )
  ( downsample ): Sequential(
23 (0): Conv2d()
  (1): BatchNorm2d()
25 )
  )
27 )
  ( layer3 ): Sequential(
29 (0): BasicBlock(
  ( conv1 )
31 ( bn1 )
  ( relu )
33 ( conv2 )
  ( bn2 )
35 ( downsample ): Sequential(
  (0): Conv2d()
37 (1): BatchNorm2d()
  )
39 )
  )
41 ( layer4 ): Sequential(
  (0): BasicBlock(
43 ( conv1 )
  ( bn1 )
45 ( relu )
  ( conv2 )
47 ( bn2 )
  ( downsample ): Sequential(
49 (0): Conv2d()
  (1): BatchNorm2d()
51 )
  )
53 )
  ( avgpool ): AdaptiveAvgPool2d()
55 ( dropout ): Dropout()
  ( fc ): Linear()
57 )
```

(2) Predictor-Architecture-2

```
1 ResNet(
  ( conv1 )
3 ( bn1 )
  ( relu )
5 ( maxpool )
  ( layer1 ): Sequential(
```

```
 7  (0):  BasicBlock(
    (conv1)
 9  (bn1)
    (relu)
11  (conv2)
    (bn2)
13  )
    )
15  (layer2):  Sequential(
    (0):  BasicBlock(
17  (conv1)
    (bn1)
19  (relu)
    (conv2)
21  (bn2)
    (downsample):  Sequential(
23  (0):  Conv2d()
    (1):  BatchNorm2d()
25  )
    )
27  )
    (avgpool)
29  (dropout)
    (fc)
31  )
```

(3) Predictor-Architecture-3

```
 1  ResNet(
    (conv1)
 3  (bn1)
    (relu)
 5  (maxpool)
    (layer1):  Sequential(
 7  (0):  BasicBlock(
    (conv1)
 9  (bn1)
    (relu)
11  (conv2)
    (bn2)
13  )
    )
15  (layer2):  Sequential(
    (0):  BasicBlock(
17  (conv1)
    (bn1)
19  (relu)
    (conv2)
21  (bn2)
    (downsample):  Sequential(
23  (0):  Conv2d()
    (1):  BatchNorm2d()
25  )
    )
27  )
    (avgpool)
29  (dropout)
    (fc)
31  )
```

(4) Predictor-Architecture-4

```
ResNet (
  (conv1)
  (bn1)
  (relu)
  (maxpool)
  (conv2)
  (bn2)
  (relu)
  (avgpool)
  (dropout)
  (fc)
)
```

## 2 ImageNet-100 Categories.

| | | | | | | | | | |
|---|---|---|---|---|---|---|---|---|---|
| n01494475 | n02095314 | n02108915 | n02177972 | n02787622 | n02971356 | n03482405 | n03899768 | n04204238 | n04509417 |
| n01644900 | n02097047 | n02111889 | n02219486 | n02791124 | n03065424 | n03496892 | n03908714 | n04235860 | n04542943 |
| n01768244 | n02097298 | n02113023 | n02229544 | n02793495 | n03124043 | n03527444 | n03977966 | n04251144 | n04548280 |
| n01770393 | n02099712 | n02113799 | n02443114 | n02814860 | n03180011 | n03544143 | n03982430 | n04258138 | n04554684 |
| n01775062 | n02101556 | n02113978 | n02494079 | n02815834 | n03201208 | n03584829 | n04041544 | n04265275 | n04590129 |
| n01797886 | n02102177 | n02115913 | n02504013 | n02825657 | n03216828 | n03598930 | n04049303 | n04270147 | n07717410 |
| n01847000 | n02105412 | n02124075 | n02606052 | n02835271 | n03372029 | n03617480 | n04081281 | n04275548 | n07920052 |
| n01910747 | n02106550 | n02128757 | n02667093 | n02837789 | n03461385 | n03637318 | n04116512 | n04417672 | n09193705 |
| n01978287 | n02108000 | n02129604 | n02687172 | n02965783 | n03476684 | n03710721 | n04118538 | n04458633 | n10565667 |
| n02085782 | n02108422 | n02165105 | n02699494 | n02966687 | n03476991 | n03804744 | n04120489 | n04487081 | n12144580 |