# OpenReview forum: "Dynamic Resolution Network "
_NeurIPS.cc/2021/Conference — NeurIPS 2021 Poster_

### Official Review · Reviewer_WvyH · 2021-07-12

**Rating:** 6
**Confidence:** 3

**Summary:**

The paper presents a method to reduce FLOP requirements by resizing images dynamically using a resolution prediction network, prepended to an image classifier network.  The resolution is selected from one of a handful of candidates, and resized before being passed to the classifier.  A training-time loss encourages smaller resolutions to be selected, effecting an accuracy vs resolution tradeoff controlled by two parameters.  Since selected resolution for inference is a one-hot categorical, the method uses Gumbel softmax to enable backpropagation during training.  A resolution-aware batchnorm layer is also described, which tracks batchnorm stats separately for each resolution.  The system performs very well, with over 40% reduction in average operations resulting in just one or two percent accuracy loss on imagenet, and small accuracy gains with 10% reduction in computation.


**Limitations And Societal Impact:**

yes

**Main Review:**

This method shows good results, and the approach of varying resolution to smallest required using a small convnet predictor is a sound approach.  The resolution-aware batchnorm is also a nice addition.  I had trouble understanding how Gumbel softmax is used: whether one-hot or mixtures are used in the training-time forward pass, and if they are one-hot, how gradients are estimated.  The description seems to indicate one-hot is used even at training time, but the referenced paper [15] describes a continuous mixture which is annealed down to one-hot over the course of training.  Currently, I don't understand how one resolution is compared to another, either within a training sample e.g. using a mixture or multiple distribution choices, or between samples e.g. using a reward or score estimator.  I think a clearer description of the mechanism that compares resolutions at training is needed.

The rest of the implementation and experiments are explained better, including the pretraining and finetuning phases, method comparisons, evaluation of parameters eta and alpha, separate batchnorms, and candidate resolutions.  It's interesting that the method performs slightly better than baseline with slightly lower FLOP count; it appears the multi-resolution training creates an effect similar to scaling data augmentation, though slightly different because images are not rescaled back to fixed size.

Overall this is a nice idea with good results.  But I think the explanation of how the forward pass sampling and backprop gradients are estimated should be clearer.




Detailed comments and questions:


- better describe Gumbel-Softmax use.  In [15], z is approximated with a continuous sample y, which is the Gumbel-Softmax, e.g. fig 2 in [15] shows y is continuous.  In this case, that would make h no longer one-hot --- how is this handled?

- [15] also use a temperature annealing schedule for tau to move from continuous to near-one-hot.  Is there an annealing schedule for tau here?


- if h is one-hot during training, how are the gradients estimated?

- X^\hat = sum_j X_j :  Since the X_j are different shapes for different resolutions, what does the sum mean here?



- what is the architecture of the resolution predictor R?  in particular, can the few conv layers used have relatively large stride? (or does increasing stride allow a resolution predictor with fewer flops, it seems fewer patch samples of the image may be needed to guess a good resolution compared to the classification net)?  l.255 mentions "4 stage residual net"


- largest resolution is 224 in these experiments --- what happens if this extended to be larger, up to 320 or even 512?  Even if this means upsampling the image, this will still change the network RF relative to object sizes.  Are the larger resolutions used when useful?  At the more extreme end, are candidate resolutions that are too large relative the network structure not selected?


- "fueled to":  is written several times, seems to be a typo.  not sure what was intended, probably "fed to" or possibly "funneled to"

- l.73: entired -> entire

- l.263: Tabel -> Table


**Time Spent Reviewing:**

3

---

> ### Author Response · Authors · 2021-08-10
> **Response to Reviewer WvyH**
>
> Thanks to the reviewer for the valuable comments.
>
> **Q1.** better describe Gumbel-Softmax use…how is this handled?
> **A1.** Thanks for the nice suggestion. Following the previous works [28], the straight-through version (fig-2(4) in [15]) of Gumbel-Max trick is used to allow for the propagation of gradients through the discrete decision. During training, for the forward pass, we get discrete samples from the argmax operation in Eq. 9, and for the backward pass we compute the gradient of the softmax relaxation in Eq. 11. Since softmax is differentiable and $g_j$ is independent noise, we can propagate gradients to the probabilities for the resolution candidates. This is the way how the one-hot is used at training time.
>
> $h =  {onehot}[(\underset{j}{\arg\max}(\log{p_{r_j}}+g_j)] $ (9)
>
> $h_j = \frac{\exp{(\log{p_{r_j}}+g_j)/\tau}} {\sum _{j=1}^{m} \exp(\log{p _{r_j}}+g_j/\tau)} $ (11)
>
> **Q2.** [15] also use a temperature annealing schedule for…annealing schedule for tau here?
> **A2.** In this paper, we directly set the temperature tau to 1 as the implementation of nn.functional.gumbel_softmax in PyTorch. We find that it does not help to the performance to anneal the softmax temperature.
>
> **Q3.** if h is one-hot during training, how are the gradients estimated?
> **A3.** The straight-through version (fig-2(4) in [15]) of Gumbel-Max trick is used to approximate the gradient of Eq. 9. During training, $h$ is one-hot in forward pass and we compute the gradient of the softmax relaxation (Eq. 11) for backward propagation.
>
> **Q4.** X^\hat = sum_j X_j: Since the X_j are different shapes for different resolutions, what does the sum mean here?
> **A4.** The Eq. 4 ($\hat{X}=\sum_{j=1}^{m}h_jX_{r_j}$) is misleading. In practical implementation, we first obtain the final prediction of each resolution: $y_{r_j}=\mathcal{F}(X_{r_j})$, and then sum them up with $h$: $\hat{y}=\sum_{j=1}^{m}h_jX_{r_j}$. We will improve the description in the final version.
>
> **Q5.** what is the architecture of the resolution predictor R?... l.255mentions "4 stage residual net"
> **A5.** We utilize the basic block of resnet to construct the predictor R, i.e., 4 basic blocks are stacked to form the predictor network in our paper. We will add more details about the predictor in the final version.
>
> Thanks for the suggestion to reduce the complexity of the predictor. Here we build 4 predictor architectures with fewer parameters and FLOPs. The details of the predictors are shown in the end of the comment.
>
> (1)	Predictor-Architecture-1: The original predictor in our paper. The parameters of the first convolution are Conv2d (3, 64, kernel_size=(7, 7), stride=(2, 2), padding=(3, 3), bias=False).
>
> (2)	Predictor-Architecture-2: We reduce the blocks of the predictor in (1) to construct a new predictor. We retain only 2 blocks.
>
> (3)	Predictor-Architecture-3: We increase the stride of the first convolution in (2) to 4. Thus, the parameters of the first convolution are Conv2d (3, 64, kernel_size=(7, 7), stride=(4, 4), padding=(3, 3), bias=False).
>
> (4)	Predictor-Architecture-4: We construct the predictor with only two convolutions.
>
> We compare the performance of different predictors as follows. We can see that the predictor with fewer flops leads to slight accuracy degradation.
>
> | Predictor | Predictor FLOPs | Total FLOPs | Acc |
> |----|----|----|----|
> |(1)| 0.29G | 3.35G | 82.5% |
> |(2)| 0.17G | 3.23G | 82.0% |
> |(3)| 0.04G | 3.10G | 82.0% |
> |(4)| 0.09G | 3.15G | 81.4% |
>
> **Q6.** largest resolution is 224 in these experiments…that are too large relative the network structure not selected?
> **A6.** We perform new experiments in ImageNet-100 with the candidate resolutions of [320, 224, 192, 168] and [512, 224, 192, 168]. The results are shown in the following table. We can see that larger resolution will obtain a better accuracy but lead to larger FLOPs.
>
> | Resolution| FLOPs | Acc|
> |----|----|----|
> |224, 192, 168|3.4G |82.5%|
> |320, 224, 192, 168|4.9G|83.3%|
> |512, 224, 192, 168|6.0G|83.3%|
>
> Our original motivation is to reduce the FLOPs of CNN by dynamically using smaller resolutions. From the above experiments, our method can also be extended to large resolution to improve the accuracy. Besides, we can further reduce the FLOPs of [320, 224, 192, 168] and [512, 224, 192, 168] with our proposed FLOPs constraint regularization.
>
> **Q7.** "fueled to": is written several times
> **A7.** We will replace "fueled to" with ''fed to'' to make it clear in the final version.
>
> **Q8.** l.73: entired -> entire.  l.263: Tabel -> Table
> **A8.**  We will correct "entired" and "Tabel" in final version.
>
> **Reference**
> [15] Eric Jang, Shixiang Gu, and Ben Poole. Categorical reparameterization with gumbel-softmax. ICLR 2017.
> [28] Andreas Veit and Serge Belongie. Convolutional networks with adaptive inference graphs. ECCV 2018.
>
> **Details of predictor architectures in A5**
>
> (1) Predictor-Architecture-1
> ```python
> ResNet(
>       (conv1)
>       (bn1)
>       (relu)
>       (maxpool)
>       (layer1): Sequential(
>         (0): BasicBlock(
>           (conv1)
>           (bn1)
>           (relu)
>           (conv2)
>           (bn2)
>         )
>       )
>       (layer2): Sequential(
>         (0): BasicBlock(
>           (conv1)
>           (bn1)
>           (relu)
>           (conv2)
>           (bn2)
>           (downsample): Sequential(
>             (0): Conv2d()
>             (1): BatchNorm2d()
>           )
>         )
>       )
>       (layer3): Sequential(
>         (0): BasicBlock(
>           (conv1)
>           (bn1)
>           (relu)
>           (conv2)
>           (bn2)
>           (downsample): Sequential(
>             (0): Conv2d()
>             (1): BatchNorm2d()
>           )
>         )
>       )
>       (layer4): Sequential(
>         (0): BasicBlock(
>           (conv1)
>           (bn1)
>           (relu)
>           (conv2)
>           (bn2)
>           (downsample): Sequential(
>             (0): Conv2d()
>             (1): BatchNorm2d()
>           )
>         )
>       )
>       (avgpool): AdaptiveAvgPool2d()
>       (dropout): Dropout()
>       (fc): Linear()
> )
> ```
>
> (2) Predictor-Architecture-2
> ```
> ResNet(
>     (conv1)
>     (bn1)
>     (relu)
>     (maxpool)
>     (layer1): Sequential(
>       (0): BasicBlock(
>         (conv1)
>         (bn1)
>         (relu)
>         (conv2)
>         (bn2)
>       )
>     )
>     (layer2): Sequential(
>       (0): BasicBlock(
>         (conv1)
>         (bn1)
>         (relu)
>         (conv2)
>         (bn2)
>         (downsample): Sequential(
>           (0): Conv2d()
>           (1): BatchNorm2d()
>         )
>       )
>     )
>     (avgpool)
>     (dropout)
>     (fc)
>   )
> ```
> (3) Predictor-Architecture-3
> ```
> ResNet(
>     (conv1)
>     (bn1)
>     (relu)
>     (maxpool)
>     (layer1): Sequential(
>       (0): BasicBlock(
>         (conv1)
>         (bn1)
>         (relu)
>         (conv2)
>         (bn2)
>       )
>     )
>     (layer2): Sequential(
>       (0): BasicBlock(
>         (conv1)
>         (bn1)
>         (relu)
>         (conv2)
>         (bn2)
>         (downsample): Sequential(
>           (0): Conv2d()
>           (1): BatchNorm2d()
>         )
>       )
>     )
>     (avgpool)
>     (dropout)
>     (fc)
>   )
> ```
>
> (4) Predictor-Architecture-4
> ```
> ResNet (
> (conv1)
> (bn1)
> (relu)
> (maxpool)
>     (conv2)
>     (bn2)
>     (relu)
>     (avgpool)
>     (dropout)
>     (fc)
>   )
> ```

---

> > ### Comment · Reviewer_WvyH · 2021-08-27
> > **post-rebuttal**
> >
> > Thanks for your responses.  My largest concern with the paper was lack of explanation for how resolutions are compared.  This is mostly addressed by the answer A4 to how the resolutions are summed:  In fact, the final logits or softmax values are summed, according to the response.  I still think the gradient for $dL/dh$ could be more precisely explained or derived, but the explanation of where the sum occurs is quite helpful in seeing how this between-resolution comparison happens.  Given that and the answers addressing several other questions, I am raising my score to 6.

---

### Official Review · Reviewer_HjSG · 2021-07-13

**Rating:** 7
**Confidence:** 5

**Summary:**

A dynamic resolution network is proposed in this submission. There is a computationally efficient predictor which selects the suitable resolution for the large classification CNN model. The predictor generates hard decision of resolutions. The usage of Gumbel softmax trick enables the gradient to propagate to the predictor. A FLOPs loss is also proposed to control the computation budget. Experiments on ImageNet show that this method could largely reduce the FLOPs with similar accuracy or improve the accuracy with fewer FLOPs reduction.

**Limitations And Societal Impact:**

1. I think the dynamic mechanism is quite unfriendly for real-world applications. For example, if all the examples are “hard”, the computation cannot be saved. To make matters worse, the proposed predictor leads to more computation.
2. There are other detailed issues in this submission.
[1] Writing errors. For example, the resolution of the input image in most of the existing compressed networks are still fixed...(line 38). Han [5] propose to generate more feature maps...(line 91). There may be more writing errors that I do not list.
[2] The statements of ‘’FLOPs constraint regularization’’ and ‘’FLOPs loss’’ are better be unified since they represent the same thing.


**Main Review:**

1. the idea using a cheap and effective CNN network to determine the input resolution of a large model is novel and convincing. It is intuitive that the difficulties of recognizing the images are diverse.
2. Besides, predicting the difficulties of the images is much easier than recognizing them, which means a tiny network is definitely sufficient to provide a good decision.
3. DRNet achieves 1.4% accuracy improvement with 10% FLOPs reduction and 34% FLOPs reduction without degeneration of accuracy, which is a fairly good result.
4. This paper clearly demonstrates this idea with good writing.
However, there are some concerns and limitations that need to be noted, e.g., some important related works are not cited.

-------------------post rebuttal---------------------

I think my concerns are well addressed in the authors' responses, So I keep my original score of 7.

**Time Spent Reviewing:**

3 hours

---

> ### Author Response · Authors · 2021-08-10
> **Response to Reviewer HjSG**
>
> Thanks to the reviewer for the valuable comments.
>
> **Q1.** I think the dynamic mechanism is quite unfriendly for real-world…more computation.
> **A1.** The ImageNet is a large-scale dataset to simulate the realistic data distribution, which contains both “hard” and “easy” examples. In practical applications, we also meet both hard and easy examples. Thus, the model faces only hard examples can be considered as an extreme case in real-world applications. Besides, compared with the large classifier, the computational cost of the predictor is very small. For example, the FLOPs of the predictor is 0.3G and the FLOPs of the classifier is 3.1G.
>
> **Q2.** There are other detailed issues…represent the same thing.
> **A2.** Thanks for the nice suggestion. We will correct these issues in the final version.

---

### Official Review · Reviewer_S6ea · 2021-07-13

**Rating:** 6
**Confidence:** 4

**Summary:**

This manuscript proposes a dynamic resolution network that can be used for the base CNN model to dynamically change the resolution of the input images. The intuition of the proposed method lies in the fact that the difficulty of the input may differ. For the easy sample, one does not need to use the full resolution for the classification. The experiments show that the proposed DRN can reduce the computational cost a lot while keeping a similar performance and even better one.

**Limitations And Societal Impact:**

1, All the experiments are conducted using images under 224*224 resolution, it would be interesting to see how the performance will be if we use a larger resolution.
2. The accuracy with lower resolutions for some examples is even better than the model with full resolution. Is there any underlying reason for this phenomenon?
3, It seems the improvement over the flops does align well with that over the real latency as shown in fig.3 and tab.3. It would be good to provide the performance and speed trade-off for real acceleration.
4, For the training process, the base models will be first trained and then combined with the resolution selector network for fine-tuning. I’m wondering if it is possible to train the whole model from scratch?

Some minor issues:
Line121: “The first is the large classifier network with both high performance and expensive computational costs is first trained”, is the “is first trained” redundant?


**Main Review:**

The intuition of the proposed method is promising and reasonable since we do not always need to use the full resolution to make the decision. The techniques, including Resolution-aware BN and the Gumbel-softmax, to achieve the dynamic resolution are interesting and make the two networks can be trained in an end-to-end manner.  The experiments indeed show the effectiveness of the proposed method.

**Time Spent Reviewing:**

2 hours

---

> ### Author Response · Authors · 2021-08-10
> **Response to Reviewer S6ea**
>
> Thanks to the reviewer for the valuable comments.
>
> **Q1.** All the experiments are conducted using images…if we use a larger resolution.
> **A1.** Thanks for the suggestion. We perform new experiments in ImageNet-100 with larger candidate resolution, i.e., [320, 224, 192, 168] and [512, 224, 192, 168]. The results are shown in the following table. We can see that larger resolution will obtain a better accuracy but lead to larger FLOPs. Besides, we can further reduce the FLOPs of [320, 224, 192, 168] and [512, 224, 192, 168] with our proposed FLOPs constraint regularization.
>
> | Resolution| FLOPs | Acc|
> |----|----|----|
> |224, 192, 168|3.4G |82.5%|
> |320, 224, 192, 168|4.9G|83.3%|
> |512, 224, 192, 168|6.0G|83.3%|
>
> Our original motivation is to reduce the FLOPs of CNN by dynamically using smaller resolutions. From the above experiments, our method can also be extended to large resolution.
>
> **Q2.** The accuracy with lower resolutions for some examples…underlying reason for this phenomenon?
> **A2.** The predictor can choose the most suitable resolution for the classifier, which makes the lower resolution performs even better than the higher resolution. For example, the image in the following link (https://i.postimg.cc/pdC1mrM5/vis.png) is correctly predicted in low resolution but wrongly predicted in high resolution. The image with higher resolution may guide the model to focus more on details instead of global information, which leads to unsatisfactory prediction.
>
>
> **Q3.** It seems the improvement over…speed trade-off for real acceleration.
> **A3.** Thanks for the nice suggestion. We tune different $\alpha$ to obtain different DRNET. We resize the input image into 224 or 192 to obtain different ResNet50 baseline. The performance and speed (latency) trade-off for real acceleration is provided as follows.
>
> | model| FLOPs | FLOPs drop |latency| latency drop | Acc |
> |----|----|----|----|----|----|
> | ResNet50-224 baseline|4.1G |-    |0.134s |-     |76.1 |
> | ResNet50-192 baseline|3.0G |27%  |0.102s |23.7% |74.3 |
> | DRNET                |3.7G |10%  |0.133s |0.7%  |77.5 |
> | DRNET                |3.2G |22%  |0.120s |10.3% |77.0 |
> | DRNET                |2.3G |44%  |0.094s |29.9% |75.3 |
>
> **Q4.** For the training process, the base models will be first trained…train the whole model from scratch?
> **A4.** We have ever trained the whole model from scratch. However, the model performance is not satisfactory. The classifier from scratch cannot make correct predictions in the early stage, the predictor cannot be trained well by the misleading of the classifier. Thus the performance is worse than that of the model with pretraining. The comparison of the two methods is shown in the following table and figure link (https://i.postimg.cc/WbVj23Lv/acc-flops.png).
>
> |model|FLOPs|Acc|
> |-|-|-|
> |DRNet w/ pretrain|2.3|80.6|
> |DRNet w/ pretrain|2.8|81.4|
> |DRNet w/ pretrain|3.3|82.3|
> |DRNet w/o pretrain|2.1|70.4|
> |DRNet w/o pretrain|2.7|77.3|
> |DRNet w/o pretrain|2.9|77.5|
>
>
> **Q5.** Some minor issues: Line121: The first is the large…redundant?
> **A5.** We will delete“is first trained” in the final version. Thanks.

---

### Official Review · Reviewer_3WL1 · 2021-07-14

**Rating:** 7
**Confidence:** 4

**Summary:**

This submission proposes a dynamic resolution network that adaptively selects the suitable resolution for image recognition. A tiny resolution predictor is proposed to predict a smaller resolution for the input image of the large classifier. Thus the computational costs can be saved. The experimental results are good and the novelty of this paper is sufficient.

**Limitations And Societal Impact:**

1. In Figure 3, the author shows the real speed improvement. The details of evaluating inference speed should be described, especially the batch size. I think that the dynamic mechanism may not reduce latency in batched inference.
2. There are many existing works on multiple resolutions and dynamic mechanisms. For example:
[1] ELASTIC: Improving CNNs with Dynamic Scaling Policies. In CVPR, 2019.
[2] Hydranets: Specialized dynamic architectures for efficient inference. In CVPR, 2018.
[3] Resolution Switchable Networks for Runtime Efficient Image Recognition. In ECCV, 2020.
These works have not been cited in this submission.


**Main Review:**

Generally, this paper proposes an intuitive and novel method, which reduces the FLOPs and improves the accuracy. The idea of reducing the resolution based on the input image is clever and convincing. The experimental results are impressive. The DRNet could yield a 1.4% accuracy increase while reducing 10% computational costs on ImageNet benchmark. Meanwhile, the author proposes a FLOPs constraint regularization, which makes the DRNet variant reduce 34% computational costs while maintaining similar accuracy. The paper is well written. Basically, I think this submission is above the standard of acceptance in NeurIPS2021.

**Time Spent Reviewing:**

Three hours.

---

> ### Author Response · Authors · 2021-08-10
> **Response to Reviewer 3WL1**
>
> Thanks to the reviewer for the valuable comments.
>
> **Q1.** In Figure 3, the author shows the…dynamic mechanism may not reduce latency in batched inference.
>
> **A1.** In this paper, we directly set the batch size as 1 for measuring the forward time, since this setting is suitable for most real-world applications. The candidate resolutions are 224, 168, and 112. The input resolution for the resolution predictor is 128. We average the test time in ImageNet-1K val set.
>
> **Q2.** There are many existing works on multiple resolutions…
>
> **A2.** Thanks for the nice suggestion. We will cite these works in the final version. ELASTIC [1] uses different scaling policies for different instances and it learns from the data how to select the best policy. Hydranets [2] chooses different branches for different inputs by a gate and aggregates their outputs with a combiner. RS-Net [3] utilizes private BNs, shared convolutions, and fully-connected layer and to train input images with different resolutions.
>
> **Reference**
> [1] ELASTIC: Improving CNNs with Dynamic Scaling Policies. In CVPR, 2019.
> [2] Hydranets: Specialized dynamic architectures for efficient inference. In CVPR, 2018.
> [3] Resolution Switchable Networks for Runtime Efficient Image Recognition. In ECCV, 2020.

---

### Decision · Program_Chairs · 2021-09-27

**Decision:**

Accept (Poster)

**Comment:**

The manuscript has been reviewed by four experienced reviewers, all of whom, after reading the rebuttal provided by the authors, agree that the manuscript meets of the bar of NeurIPS and thus should be presented to a large audience. The AC also agrees that the proposed approach is novel, supported by sufficient emperical evaluations, and hence recommends acceptance.